# Systematic Review of Platelet-Rich Plasma for Low Back Pain

**DOI:** 10.3390/biomedicines11092404

**Published:** 2023-08-28

**Authors:** Edilson Silva Machado, Fabiano Pasqualotto Soares, Ernani Vianna de Abreu, Taís Amara da Costa de Souza, Robert Meves, Hans Grohs, Mary A. Ambach, Annu Navani, Renato Bevillaqua de Castro, Daniel Humberto Pozza, José Manuel Peixoto Caldas

**Affiliations:** 1REGENERAR—Pain Medical Center, Porto Alegre 90620-130, Brazil; 2PhD (c) Faculty of Medicine, University of Porto, 4200-135 Porto, Portugal; 3Spine Group, Hospital Ernesto Dornelles, Porto Alegre 90160-092, Brazil; 4Department of Orthopedics and Traumatology, Faculdade de Ciências Médicas da Santa Casa de São Paulo, São Paulo 01224-001, Brazilhansmed@terra.com.br (H.G.); 5San Diego Orthobiologics Medical Group, Carlsbad, CA 92011, USA; 6Le Reve Regenerative Wellness, Campbell, CA 95008, USA; 7Center for Tissue Regeneration Studies, Campinas 13076-629, Brazil; 8Department of Biomedicine, Faculty of Medicine, University of Porto, 4200-319 Porto, Portugal; dhpozza@gmail.com; 9Institute for Research and Innovation in Health and IBMC, University of Porto, 4200-135 Porto, Portugal; 10CIEG-ISCSP, University of Lisbon Camp, 1300-663 Lisboa, Portugal; 11Instituto de Saúde Pública da Universidade do Porto (ISPUP), 4050-600 Porto, Portugal

**Keywords:** platelet-rich plasma, low back pain, discogenic pain, facet pain, orthobiologics, regenerative medicine, spinal injection, epidural injection

## Abstract

Background: Low back pain (LBP) has a high economic burden and is strongly related to the degenerative process of the spine, especially in the intervertebral disc and of the facet joints. Numerous treatment modalities have been proposed for the management of LBP, and the use of platelet-rich plasma (PRP) has emerged as an innovative therapeutic option for degenerative disease of the spine. The present study aims to evaluate the efficacy of PRP injections in managing low back pain. Methods: We conducted a systematic review in accordance with the Preferred Reporting Items for Systematic Reviews and Meta-Analyses (PRISMA) recommendations, a registered at PROSPERO Systematic Reviews Platform, under number CRD42021268491. The PubMed, Web of Science, and Scopus databases were searched to identify relevant articles, along with hand searching to identify gray literature articles, with no language restrictions. Randomized clinical trials (RCTs), nonrandomized trials (NRTs), and case series (CSs) with more than 10 patients were considered eligible. The quality assessment and the risk of bias of the randomized clinical trials were evaluated using the RoB II tool. An evaluation of the description of the preparation methods was performed using an adapted version of the MIBO checklist. Results: An electronic database search resulted in 2324 articles, and after the exclusion of noneligible articles, 13 RCTs and 27 NRTs or CSs were analyzed. Of the 13 RCTs, 11 found favorable results in comparison to the control group in pain and disability, one showed no superiority to the control group, and one was discontinued because of the lack of therapeutic effect at eight-week evaluation. Description of the PRP preparation techniques were found in almost all papers. The overall risk of bias was considered high in 2 papers and low in 11. An adapted MIBO checklist showed a 72.7% compliance rate in the selected areas. Conclusions: In this systematic review, we analyzed articles from English, Spanish and Russian language, from large databases and grey literature. PRP was in general an effective and safe treatment for degenerative LPB. Positive results were found in almost studies, a small number of adverse events were related, the risk of bias of the RCTs was low. Based on the evaluation of the included studies, we graded as level II the quality of the evidence supporting the use of PRP in LBP. Large-scale, multicenter RCTs are still needed to confirm these findings.

## 1. Introduction

Musculoskeletal diseases are one of the main causes of disability, being a burden to society, with costs higher than many diseases, including direct and indirect expenses such as loss of productivity and early retirement [1,2]. Low back pain (LBP) is the most prevalent musculoskeletal disease, affecting between 20% and 50% of the adult population, and it is the leading diagnosis of years lived with disability (YLD) in the general population [3,4,5]. Degenerative processes in the spine, mainly in the intervertebral discs [6], and of the facet joint (i.e., facet arthrosis) are frequently found in LBP [7,8,9]. However, therapies that restore degenerated structures are not available in current clinical practice [10].

Numerous treatment modalities with various approaches, from cognitive-behavioral therapy, exercises, to major surgeries, have been proposed for the management of LBP. Orthobiologics, involving cells or substances related to the healing or regenerative process of human tissues can be a therapeutic option for degenerative joint disease [11]. The most studied orthobiologic agent is platelet-rich plasma (PRP), which is easily obtained through peripheral blood centrifugation [12]. The efficacy and safety of PRP use in orthopedic patients, especially those with knee arthrosis, are well supported in the literature, allowing for pain control and improvement in clinical scores [13,14].

The use of PRP in spinal pathologies is growing because of a better understanding of the physiology of the intervertebral disc and related degenerative diseases [15]. The low-grade inflammatory process in the intervertebral disc, mediated by cells such as chondrocytes in the nucleus pulposus and fibroblasts in the annulus fibrosus, leads to a slow extracellular matrix degenerative process [16]. Although experimental use of isolated growth factors can revert the degenerative process, it still has no clinical application [17]. In this context, PRP, which releases several growth factors upon activation, has shown promise in animal studies as a potential clinical option to decrease the degenerative process [18,19,20,21].

Clinical research has also been conducted, with an increasing number of published studies allowing for some systematic reviews with meta-analyses [22,23]. It has been demonstrated that the administration of orthobiologics, including PRP, in the treatment of LBP and sciatica presents favorable clinical results [24]. On the other hand, other intradiscal drugs and nonbiologic treatments did not show any clear benefits in different intradiscal therapies. Furthermore, serious adverse effects have been observed in intradiscal therapy with corticosteroids, highlighting the importance of exploring other options with the potential for long-lasting relief and repair, such as orthobiologics [25].

### 1.1. Manifestations of Degenerative Disease

Degenerative disease of spine disease may include internal disc disruption, disc herniation, facet arthropathy, muscle atrophy, and spinal stenosis. Such alterations can manifest alone or in combination: discogenic low back pain, muscle atrophy, facet arthropathy, and disc herniation.

Internal disc disruption: One of the most common causes of low back pain, a painful disk, without herniation, is a consequence of internal ruptures leading to vascularized granulation tissue invasion with extensive innervation, in an attempt to heal [26,27,28]. Magnetic resonance imaging can show, in some patients, loss of T2 signal (dehydration) and/or hyperintense signal in the posterior disc from fissures called HIZ, or high-intensity zone [29]. In the vast majority of discogenic pain cases, the degenerative alterations are Pfirrmann grade II or III [30]. Modic Type I changes have also been correlated with pain and positive discography alterations [31]. The use of provocative discography can be useful, although its use should be weighed against the possible risks of this more involved procedure [32,33,34,35].Disc herniation is a consequence of hydration losses and capacity to absorb and distribute compressive loads, making it susceptible to internal fissures that can progress to complete ruptures and leak of disc tissues into the spinal canal [36]. This herniation can lead to radiculopathy, either through direct mechanical compression of the nerve tissue or through an inflammatory reaction triggered by the release of various cytokines, particularly TNF alpha and IL-6 [37].Facet joint pain/syndrome is a prevalent complaint among individuals experiencing low back pain. Facet joints are diarthrodial joints encompassing a joint capsule and synovial membrane, with surfaces covered by cartilage. The oblique orientation of these joints contributes to their resistance against shear forces and limitation of rotational movement [38]. It was demonstrated that in a degenerated disc, axial compression is not adequately absorbed, resulting in the transmission of this load to the facet joints. Consequently, the facets experience an increase in mechanical demand by four to eight times their original capacity. This heightened load, coupled with increased instability, leads to joint injuries, and initiates the degenerative process within the facets [39,40].The atrophy of the paravertebral musculature has been associated with low back pain [41]. Animal studies and magnetic resonance images in humans with discogenic low back pain found a causal relationship between fatty infiltration of the musculature and discogenic pain [42]. Muscle atrophy and fatty degeneration are commonly observed in individuals with chronic low back pain, indicating the vital role played by the paraspinal muscles in maintaining lumbar spine stability [43].

### 1.2. Treatment Options

#### 1.2.1. Medication and Physical Activity

The treatment approach for chronic low back pain includes NSAIDs, antidepressants, exercise therapy, and psychosocial interventions. When other analgesics are not enough, opioids can be considered if the benefits outweigh the risks [44,45].

Exercise demonstrates improvements in pain reduction and functional limitations compared to other conservative treatment options [46]. However, there is no consensus on which specific therapy is the most effective. While one study found greater benefits in Pilates therapy and strength training and fewer benefits in stretching exercises [47], there is still no conclusive agreement on the best approach. Various modalities of techniques and exercises, such as walking, yoga, tai chi, and stretching, are proposed, and the choice among them can be based on the individual’s personal preference, since there is no evidence of one particular approach being superior to the others [48].

#### 1.2.2. Interventional Measures

When standard pharmacological measures and physical therapies fail to effectively manage symptoms, it is advisable to consult with a specialist for further evaluation. At this stage, targeted treatments can be considered. Interventional procedures aimed at pain control include thermoablation [33] or orthobiologic injections for discogenic pain [34]. In the case of facet joint pain, options include injections with anesthetics and/or corticosteroids, as well as thermoablation of the capsular nerves or middle branch nerves [40]. Additionally, epidural steroid injections are commonly used for managing radiculopathy or spinal stenosis.

#### 1.2.3. Surgery

The main reason to indicate surgery is to approach the damaged nerve and/or other anatomic structures affected by the degenerative process. However, the surgical approach, per se, is still not able to delay or reverse the underlying degeneration itself. Open discectomy, accounting for almost 70% of surgical cases, is the most common surgery for advanced low back pain. The remaining procedures include endoscopic discectomy, laminectomy, and, less common, fusion or nucleolysis. Reoperation rates vary according to the case and the type of surgery, with laminectomy having a higher reintervention rate of approximately 19%. Comparing open discectomy, the reoperation rate was higher for laminectomy at 3 months, while the other surgical methods had similar rates [49]. In a meta-analysis comparing more complex surgical techniques, like arthroplasty, anterior, and posterior fusion, an average of 1.13 complications per case was reported [50].

#### 1.2.4. Regenerative Medicine

Current standard treatments focus on the degenerative processes without directly impacting their progression. However, in line with the principles of regenerative medicine, which involve replacing, repairing, or regenerating human cells, tissues, or organs to restore normal functions [51], researchers have been exploring alternatives to control or reverse the degeneration of spinal structures. Understanding healing mechanisms has led to therapies targeting degenerative processes, with promising results in degenerative disc and joint diseases [52].

Orthobiologics, which tap into the repair and regeneration potential inherent in the body’s own cells, have gained prominence in both research and clinical practice. These therapies employ cells or substances derived from the human body to enhance the natural healing or regeneration process in muscles, tendons, and cartilage. Platelet-rich plasma (PRP), which is an orthobiologic obtained from an individual’s own peripheral blood, is subjected to centrifugation to separate its components based on their specific weights. Growth factors released by activated platelets, such as PDGF, TGF, VEGF, IGF, and EGF, play vital roles in tissue repair and healing, influencing processes like hemostasis, inflammation, proliferation, and remodeling. These growth factors can attract mesenchymal cells and stimulate myoblast, chondroblast, and osteoblast differentiation [53].

The PRP offers a simple, safe, and cost-effective procedure, but there is no consensus on the optimal preparation protocol. High-level evidence studies across various specialties demonstrates favorable results for PRP in orthopedic pathologies, with outcomes superior to the control group in 61% of cases [54]. The most highly cited articles on PRP show level 1 evidence and positive outcomes, with a growing number of clinical, in vitro, and animal studies [55].

#### 1.2.5. PRP in Degenerative Spine Disease

It was demonstrated that PRP has regenerative effects with the ability to stimulate cell activity and increase collagen and proteoglycan production [19,20]. In a series of patients with discogenic low back pain, preliminary results from intradiscal PRP therapy showed both the safety of the technique and symptom improvement over a six-month follow-up period [56]. A subsequent evaluation, with a follow-up period of 5.9 years, demonstrated sustained pain and disability improvement in over 90% of the patients [57]. Additionally, PRP can significantly improve low back pain, disability scores, and walking ability scores in short-term treatment with no clinically significant adverse events [58].

In experimental animal models of disc degeneration, PRP, combined or not with bone marrow cells, showed inhibitory effects on degeneration and significantly restored disc properties. It was also demonstrated that PRP has an anabolic effect on disc cells, promoting increased chondrocyte activity, collagen production, and proteoglycan synthesis [59,60]. Clinical studies have reported positive outcomes, safety, and superiority over placebo of intradiscal PRP treatment, including improvements in pain and muscular atrophy [61].

Evidence level considering the risk of bias, indirectness, and imprecision should be evaluated for PRP but also for the other proposed treatments of low back pain due to disc degeneration [62]. PRP also demonstrated efficacy and safety for lumbar facet pain when compared to local anesthetics and corticosteroids injected intra-articularly [63]. Epidural PRP injections have demonstrated long-term safety and effectiveness, with platelet lysate injections leading to pain reduction and functional improvement [64]. In terms of pain and disability, better results were demonstrated for PRP rich in leukocytes for patients experiencing complex low back pain due to degenerative changes when compared to pharmacological treatment using corticosteroids [60].

#### 1.2.6. Study Aims

Based on previous reports, the primary objective of the present study was to assess the effectiveness of PRP injections in the management of low back pain. Additionally, the study aimed to evaluate the different PRP preparation protocols and characteristics of the PRP product. To achieve these objectives, a systematic review was conducted.

## 2. Materials and Methods

### 2.1. Literature Search and Search Strategy

The research was carried out in accordance with the Preferred Reporting Items for Systematic Reviews and Meta-Analyses (PRISMA) methodology [65] (see PRISMA checklist at Appendix A), registered on the PROSPERO Systematic Reviews Platform (https://www.crd.york.ac.uk/prospero/, accessed on 1 February 2023), under number CRD42021268491, URL accessed on 11 November 2022.

The first part of the study consisted of an exhaustive search and definition of the descriptors (MeSH terms) by two researchers (E. M. and M. B.). Descriptor terms were defined independently and validated by consensus. 

The descriptors were divided into blocks ((1) low back pain, (2) intervention, and (3) product) and then combined. The search strategy is detailed in Appendix B. The second stage was carried out by searching the following databases: PubMed, Scopus, and Web of Science. The search was carried out from 7 December to 30 December 2021, by two researchers (R. B. P. and C. M. D. A.). Two independent groups of researchers screened the articles, with an initial reading of the title and abstract. Differences were resolved through a consensus meeting between the two groups. After the final selection of articles, they were distributed for data extraction.

The references obtained through search engines in the cited sources had duplicate articles excluded and were evaluated by two independent reviewers (P. P. and L. S.), according to the inclusion and exclusion criteria. After comparing the list of inclusions, the differences were discussed with the project coordinator in a consensus meeting.

### 2.2. Eligibility Criteria

Randomized clinical trials, nonrandomized trials, and case series with more than 10 patients were considered eligible. The following were excluded: animal or in vitro studies, review studies, systematic reviews, meta-analyses, editorials, letters to the editor, case series with less than 10 patients, and case reports. A summary of selection criteria according to the PICO elements is shown in Table 1.

### 2.3. Data Extraction

Four independent reviewers (E. M., H. G., P. P., and L. S.) performed the data extraction independently, and disagreements were resolved in a consensus meeting. General characteristics of the studies were collected, such as: authors, year of publication, study design, sample size, intervention, target, follow-up time, adverse events, description of the preparation, and main conclusions.

### 2.4. Quality Assessment and Risk of Bias

The quality assessment and the risk of bias of the randomized clinical trials were evaluated using the RoB II tool [66].

In addition to the articles selected in the systematic search, articles obtained from other sources (manual search and bibliographic references) that, by common agreement, were considered relevant and that met the inclusion criteria in the study were included.

The evaluation of the description of the preparation method and specific aspects of studies with orthobiologics was performed using a checklist adapted from Murray et al. [67].

The analysis of the evidence was performed based on the best evidence synthesis according to the ASIPP-modified approach to grading of evidence (Table 2) [24].

## 3. Results

The search engine resulted in 2324 articles. The manual search of articles presented in bibliographic citations, congress annals, and nonindexed literature added up to 13 more titles. After excluding duplicate titles (*n* = 202), the titles and abstracts of the rest were checked, with 2107 articles excluded for not meeting the inclusion criteria. A total of 40 articles on the use of PRP in the lumbar spine were selected for reading and analysis, including 13 randomized clinical trials and 27 case series (Figure 1: PRISMA flowchart).

The 13 randomized controlled trials (RCTs) included a total of 914 patients, with 7 studies focusing on intradiscal injections of PRP (427 patients) [58,68,69,70,71,72,73], 3 on epidural injections (267 patients) [74,75,76], 2 on facet injections (190 patients) [63,77], and 1 on muscle and ligament injections (30 patients) [78]. One RTC study was still in the peer-review process at the time of analysis and was included as a personal communication to the author [73]. Additionally, the case series involved a total of 1759 patients, with studies focusing on intradiscal (160 patients) [34,56,79,80,81,82,83], epidural (810 patients) [64,84,85,86,87,88], facet injection (109 patients) [89,90,91], and lumbar muscles (171 patients) [61,92] or simultaneous application in multiple targets (509 patients) [10,93,94,95,96,97,98,99,100]. The RCTs compared the effect of PRP with saline, steroids, contrast agent, hyaluronic acid, or ozone.

Table 3 presents the main findings of the RCTs on intradiscal injection of PRP. Out of the seven studies, one was discontinued after 8 weeks [70], one lacked detailed methodology with a high risk of bias [72], and another was still in the process of peer review [73].

Table 4 shows the main findings of the RCT of epidural, facet, or muscle injection of PRP. The Figure 2 shows the results of the analysis of the intradiscal PRP RCTs based on the modified MIBO checklist. The Figure 3 shows the results of analysis of the PRP in other sites based on the modified MIBO checklist.

In the case series group, a total of 27 articles were found. Table 5 presents 18 papers focusing on specific targets, including the disc (*n* = 7) [34,56,79,80,81,82,83], epidural space (*n* = 6) [64,84,85,86,87,88], facet joint (*n* = 3) [89,90,91], and paravertebral muscle (*n* = 2) [61,92]. On the other hand, Table 6 includes nine papers that explored multitarget applications, involving the disc, facet, epidural space, and/or muscles simultaneously [10,93,94,95,96,97,98,99,100]. The combined number of patients in both tables amounts to 1759 individuals. The most frequent primary outcome assessed across these studies was pain intensity that was measured by VAS or NRS. Functional outcomes were evaluated in some studies using ODI, Roland–Morris or SF-36.

### 3.1. Outcome Measure Tools—Pain and Disability

In the RCTs group, pain intensity was measured using the visual analogue score (VAS), numeric rating score (NRS), or Lattinen pain score. Compared with the baseline values, all articles demonstrated a decrease in pain on the evaluations of 26 or 52 or 60 weeks of follow-up. When compared with the control group, articles showed similar or superior results (Table 7).

Tuakly et al. [68] showed, in a participant-reported NRS best pain score, a significant difference over 8 weeks compared with the control group (*p* = 0.02). Comparing the baseline values of pain, and the 12 months evaluation, there was a decrease of 2.12 points in the NRS (to 7.98 from 5.85).

To evaluate functional outcomes, five studies used the Oswestry Disability Index (ODI), four studies used the Roland–Morris Disability Questionnaire (RMQ), two used the SF-36, and one used the Lattinen (Table 8).

### 3.2. PRP Technique

A description of the PRP technique was found in almost all papers, with variable details. A commercial kit was used in 16 studies. A manual preparation was conducted in 20 studies. Only three papers did not show any information concerning the preparation methods. The PRP characteristics are shown in Table 3, Table 4, Table 5 and Table 6.

#### Risk of Bias

The quality assessment and the risk of bias of the RCT studies were analyzed using the RoB II tool (Table 9).

The other articles selected consisted of case series reporting the results of the use of PRP in the intradiscal, epidural, or facet route alone or on several targets simultaneously. The list of these articles can be found in Table 4 and Table 5, respectively.

An evaluation of the studies was conducted based on an adapted version of the MIBO checklist [67], which consisted of eleven items (Figure 2 and Figure 3). The purpose of this evaluation was to assess the adequacy of study descriptions, including details of the study population, PRP preparation, and acquisition methods, intervention techniques, and obtained results. The overall compliance rate across the various selected areas was found to be 72.73%. Partial descriptions were observed in 15.15% of the items, while 12.12% of the items were either not described or inadequately described. These results indicate a satisfactory adherence to the protocol proposed by the AAOS expert group in 2017, which is promising for the quality and reproducibility of the studies.

## 4. Discussion

This systematic literature review found 13 RCTs and 27 non-RCTs on the use of PRP for low back pain. The etiology of the pain and the sites of application were varied, with the majority of the studies on the treatment of discogenic pain with intradiscal biologics. Evidence has consistently demonstrated that PRP therapy offers a less invasive and safe alternative for the treatment of chronic low back pain. Notably, the use of PRP has shown significant benefits without substantial side effects or complications. This highlights the potential of PRP as a promising therapeutic approach that not only addresses chronic low back pain effectively but also minimizes the risks associated with more invasive treatment options.

### 4.1. Intradiscal Injections

In comparison to placebo (intradiscal contrast), PRP injections for Low Back Pain demonstrated significant improvements in functional outcomes, pain relief, and patient satisfaction. Importantly, no complications such as progressive disc herniation or neurological deficits, were observed in the treated patients [68]. Disc space infection was reported in only one study [71]. Remarkably, those who received PRP injections maintained their pain relief and improved function even several years after the injection [101]. Multidimensional pain evaluation, including physical, functional, and emotional assessments, revealed significant clinical improvements. Even after many years of treatment, the majority of patients expressed satisfaction and reported sustained pain relief and functional improvement following intradiscal PRP injections. However, during the follow-up period, a subset of subjects who underwent surgery were classified as treatment failures.

PRP also demonstrated significantly improved disability scores and better walking ability scores when compared to corticosteroids in patients with discogenic low back pain [58]. No clinically important adverse events were observed. The same group previously published studies on the use of PRP in vitro, intradiscal studies in animals, and nonrandomized case series, demonstrating positive clinical outcomes in patients over 5 years of follow-up [57]. Intradiscal PRP also demonstrated to be safe and effective, with fewer side effects than ozone therapy in discogenic pain patients followed up for one year [72].

Another RCT with 89 patients suffering from discogenic low back pain compared PRP with saline solution along with kefazol. The one-year follow-up evaluation showed similar results between the groups, indicating no superiority of PRP over saline solution. However, the study had some limitations, including a low volume of blood draw with no reported platelet counts, the control group receiving an antibiotic injection, and exclusion of Modic changes, which may have affected the findings [71]. In this context, the protocol of PRP must be carefully chosen in order to obtain the maximum benefits.

Comparisons among different concentrations of PRP were performed: average concentration of 5× the baseline value and a new series using ultra-concentrated PRP rich in leukocytes (LR-PRP) with an average of 10× more platelets than the baseline level. The results demonstrate that the initial pain and disability were higher for the new higher concentration. However, the percentage of improvement in pain, satisfaction, and disability was more pronounced in the end for the higher concentrations [82], which is in accordance with other clinical studies that also used higher PRP concentrations [81]. The last study emphasized the existence of several concentration protocols, even with standardized methods [81]. In addition to the concentration, the composition is also important, and it was demonstrated that platelet-rich fibrin (PRF) is also safe and effective for discogenic low back pain. Furthermore, the comparative clinical evaluation with PRP suggested better results in the PRF group [69].

### 4.2. Epidural Injections

The epidural route is commonly used for administering medications to treat low back pain [102]. Among interventional procedures, the injection of anesthetics with corticosteroids is widely employed, although its superiority over other injectates is subject to debate. A systematic review with meta-analysis revealed favorable outcomes with the use of blocks containing only anesthetics compared to blocks with anesthetics and corticosteroids [103].

In our literature review, we identified three randomized controlled trials (RCTs) and six case series that utilized epidural PRP injections. Additionally, there were nine papers describing multitarget injections that included epidural, disc, and facet injections.

One RCT conducted by Ruiz-Lopez & Tsai (2020) investigated the use of PRP administered via the caudal route in 50 patients with low back pain, demonstrating significant improvement in pain and disability. The study also found PRP to be superior to corticosteroids after a 6-month follow-up period [74].

Núñez et al. (2021) published another RCT comparing the use of PRP and translaminar corticosteroids in 93 patients with low back pain. In addition to significant clinical improvement, the PRP group reported fewer adverse reactions (*n* = 27) compared to the corticosteroid group (*n* = 103) [76].

An RCT involving 124 patients with lumbar radiculopathy examined the epidural transforaminal access for PRP injection. At the 12-month follow-up, both the PRP and corticosteroid groups showed similar clinical outcomes in terms of pain and disability, with no reported adverse events [75].

The use of epidural PRP for cervical and lumbar disc herniation significantly improved patient outcomes. The initial case series showed that 40% of the patients experienced complete symptom relief, while the follow-up study demonstrated a substantial decrease in pain levels and improved functional outcomes. These findings highlight the potential efficacy of epidural PRP as a treatment option, with positive results were observed in terms of pain reduction and decreased reliance on opioids [86].

Epidural platelet lysate administration was suggested as a viable alternative to corticosteroids because of its benefits in pain and function. There is a lack of clear evidence regarding the benefits of corticosteroids in improving function, reducing disability, or avoiding surgery. Additionally, several randomized trials have shown the lack of superiority of steroids over placebo and highlighted the numerous adverse effects associated with corticosteroid use, impacting various bodily systems [103].

### 4.3. Facet Injections

Facet joint pain is a common condition observed in clinical practice, particularly in the lumbar spine. The overload and degeneration of the facet joints can be attributed to the failure of the shock absorber mechanism of the degenerated disc [39]. In addition to two randomized controlled trials (RCTs) and three case series, we found nine multitarget studies that included injections targeting facets, discs, and the epidural space.

PRP use in 49 facet syndrome patients resulted in significant pain reduction and improved functionality after 18 months. No adverse reactions were reported, confirming the high effectiveness of PRP for managing facet joint pain [90] The same research group conducted an RCT with 144 patients diagnosed with facet joint pain, comparing PRP and hyaluronic acid. After an average 18-month follow-up, both groups showed similar results, but the PRP group exhibited superior clinical improvement and higher patient satisfaction. Complications such as infection, subcutaneous hematoma, and worsening of symptoms occurred in both groups but were effectively managed. Overall, patients achieved positive clinical outcomes in the final evaluation [77].

The two-step centrifugation method PRP is a good alternative to approach patients with lumbar facet syndrome. This modality of treatment, by intra-articular injections, demonstrated being safe and effective without complications, when compared to local anesthetics and corticosteroids. The corticosteroid group initially had higher satisfaction and success rates, but these declined after 6 months. In contrast, the PRP group demonstrated continued improvement over time [63].

### 4.4. Other Spinal Target Sites

Atrophy of paraspinal muscles can lead to increased stress on the facets and discs, contributing to a cycle of pain and degeneration, since these muscles play a crucial role stabilizing the spine. The weekly PRP injections along with physiotherapy and walking in the lower back muscles of 115 patients allowed an overall success rate of 71% after one-year. Furthermore, post-procedure MRIs showed an improvement in pre-existing multifidus muscle atrophy, and patient satisfaction reached 87.8% [61].

In a study involving 30 patients with chronic nonspecific low back pain, PRP injections were administered in the lumbar ligaments, muscles, and fascias [78]. After six-month follow-up, all pain and disability assessments favored PRP. It was also highlighted the synergistic effect of PRP and prolotherapy in strengthening the fascia and ligaments in the lumbosacral region. However, it is important to note that the study had a limitation of lacking ultrasound guidance during the injections. The use of ultrasound imaging can potentially enhance the effectiveness of these procedures by visualizing the fascia and ligaments and ensuring accurate delivery of the medication to the intended area.

### 4.5. Compliance According to the MIBO Assessment Checklist

The evaluation of the studies conducted based on an adapted version of the MIBO checklist [67] consisted of eleven items. The adequacy of study descriptions, including details of the study population, PRP preparation and acquisition methods, intervention techniques, and obtained results were evaluated. Our revision showed a compliance rate of 72.73%. Partial descriptions were observed in 15.15% of the items, while 12.12% of the items were either not described or inadequately described. In a previous systematic review involving 19 studies and 1005 patients, the use of PRP in shoulder procedures was evaluated. The review found that 58.5% of the 47 checklist items from the MIBO checklist were reported across all studies [104]. It is important to note that the original protocol comprised nearly five dozen items to be reviewed. In our review, we opted for a simplified version with eleven items, which provides a more practical and objective assessment approach (see Figure 2 and Figure 3 for details).

Overall, the evaluation of the studies based on the adapted MIBO checklist highlights the efforts made to adequately describe the research protocols and findings. However, there is still room for improvement in terms of providing complete descriptions for all relevant aspects of the studies, ensuring transparency, and facilitating reproducibility in future research endeavors.

The studies reviewed in the literature described a wide range of PRP preparation methods, including both commercial kits and different other techniques. The resulting PRP products also exhibit variability, with LR-PRP (leukocyte-rich PRP) being the most frequently discussed and supported for use in chronic low back pain. The diverse preparation methods and variations in the final PRP product, such as platelet concentration and presence or absence of leukocytes, can lead to variations in the biological effects of PRP. Different types of PRP concentrates can be produced with varying characteristics based on the concentration of its components, each exerting a distinct biological effect. This analysis provides valuable data to determine the ideal type of PRP for specific pathologies. For example, gluteal tendinopathy, characterized by atrophic-ischemic lesions, may benefit from leukocyte-rich PRP treatment [105].

In the context of intra-articular applications, in vitro studies have suggested that leukocytes could have a detrimental effect on joint cartilage [106]. On the other hand, a systematic review [107], which included 32 studies with evidence levels ranging from 1 to 4, found similar functional and clinical results between leukocyte-rich and leukocyte-poor PRP preparations in knee treatments. Nevertheless, the incidence of adverse reactions was higher in patients receiving LR-PRP, with an odds ratio of 1.64. Another review [108] demonstrated that both leukocyte-poor and leukocyte-rich PRP can benefit patients. In the treatment of lateral elbow epicondylitis, multiple studies and systematic reviews have already shown the effectiveness of PRP [109]. Two systematic reviews did not find differences in the final outcomes between PRP rich or poor in leukocytes. However, one of the previous reviews found a slightly higher incidence of side effects with leukocyte-rich injections [110,111].

Apart from leukocytes, macrophages also play an important role in LBP inflammation including innate immunity, tissue repair, and remodeling. Macrophages can promote tissue damage and inflammation (called M1) or support tissue remodeling and suppress inflammation (called M2) [112,113]. M1 were found in osteoarthritic joints contributing to disease progression, suggesting they can be targeted for treatment. Although many studies have reported PRP’s ability to suppress inflammation and induce articular cartilage repair, the effect of PRP on macrophage phenotypes has not been fully explored.

The composition and preparation protocols of PRP allows for a huge variation of mechanisms and different clinical implications. PRP can modulate inflammation and consequently influence the M1 to M2 phenotype transition of macrophages, being influenced by the presence or absence of leukocytes. It was demonstrated that both leukocyte-rich and leukocyte-poor PRP facilitated the recruitment of M1 macrophages to the injury site, contributing to early-stage tissue repair. However, only leukocyte-poor PRP elicited the activation of M2 macrophages [114]. Platelet-rich concentrates, with increased concentrations of growth factors, like TGF-β, PDGF, and IGF, have shown positive effects on nerve healing, as well as the ability to modify macrophage phenotypes and inhibit inflammation through the release of bioactive molecules like lipoxin [115,116]. Although the potential deleterious effects of leukocytes on PRP composition may increase M1 expression, further research is needed to draw definitive conclusions. Currently, scientific evidence points to a beneficial effect of PRP on the polarization of M1 macrophages into M2 macrophages [117].

The best type of PRP protocol preparation, the most appropriate type of PRP for each specific anatomical location and the different combinations of PRP that can be used in different areas, according to the patient needs, are part of the precision medicine and treatment individualization [118]. This approach acknowledges the importance of personalizing treatments based on the unique characteristics and requirements of each patient.

### 4.6. Multitarget Therapy

One important finding in this literature review was the identification of a case series where the authors advocate for the application of injections in multiple sites rather than targeting a single area (as shown in Table 5). It is well established that there is a strong relationship between disc degenerative disease and facet arthrosis [119]. Furthermore, it is common to observe simultaneous degenerative lesions in real-world patients. Therefore, the rationale behind treating patients with multiple targets is justifiable.

The degenerative process in spinal disease exhibits distinct characteristics, affecting various biological aspects, thereby requiring an individualized evaluation for each patient. Understanding the degenerative process in the spine and recognizing the interconnectedness of different spinal structures implies that most patients with chronic low back pain do not experience pain originating from a single source. The aforementioned [10] case series of chronic low back pain, focused on multitarget PRP injections and demonstrated that around 82% of the patients’ exhibited changes in two or more sites, with facet arthrosis and disc degeneration being the most frequently observed. Since LBP disease process presents a multifactorial nature, involving many anatomical sites, it is extremely difficult to define its cause, even by using complementary exams. Sometimes multiple abnormalities, such as facet joint arthropathy, intervertebral disc pathology, spinal canal stenosis, and paravertebral muscle atrophy can be detected through MRI, demonstrating the importance of using image.

#### Evidence Analysis

Evidence analysis was conducted using the Manchikanti approach adapted by Navani (see Table 1).

For intradiscal injections, our search identified seven RCTs (Table 2) [58,68,69,70,71,72,73] (one in peer-review process), with four having a low risk of bias [58,68,71,73] and one with a high risk [72]. Additionally, we found 16 nonrandomized studies or case series (Table 4 and Table 5), the majority of which were of moderate quality. Based on a qualitative assessment, there is grade II evidence supporting the use of intradiscal injection of platelet-rich plasma (PRP) for lumbar discogenic pain.

Regarding epidural injections, our search yielded three RCTs (Table 3) [74,75,76], with two classified as having a low risk of bias [74,75] and one with a high risk [76]. We also identified 15 nonrandomized studies or case series (Table 4 and Table 5), mostly of moderate quality. The qualitative analysis suggests a grade II evidence level for epidural injection of PRP in the management of low back pain.

In the case of facet injections, our search identified 2 RCTs (Table 3) [77,78] with a low risk of bias, along with 12 nonrandomized studies or case series (Table 4 and Table 5), again mostly of moderate quality. According to the qualitative approach, there is grade II evidence supporting the use of facet injection of PRP for low back pain.

Despite significant advancements in technology in recent years, the intricate methodology of clinical studies has hindered the translation of research findings into practical applications. The use of platelet concentrates in surgical practice was first reported in the late 1990s [120], while the use of PRP for spinal pathologies has been described for over a decade [94]. However, in more than 10 years, only 13 RCTs with limited patient samples and a few dozen case series have been conducted, predominantly derived from the authors’ clinical practices. While RCTs have traditionally been considered the gold standard for generating evidence, their execution is not always feasible. Barriers such as cost, patient selection, and the time required to conduct and conclude a study create a significant gap between evidence production and real-world implementation. Krittanawong [121] noted that over 30,000 clinical trials related to cardiovascular disease in clinicaltrials.gov either remain incomplete or have not reported results in peer-reviewed publications because of challenges like limited statistical power (difficulty in recruiting suitable candidates), inadequate follow-up periods, heterogeneous study populations with unaccounted inter-individual variabilities, irrelevant adverse events, or publication bias.

In recent years, new models of clinical trial design have emerged, including master observational trials (MOTs), umbrella studies, platform studies, and basket studies [122]. One of the major criticisms of traditional clinical trials is their limited representation of real-world populations, making their findings less applicable to actual patients. The principles of personalized medicine or precision medicine, which aim to tailor treatments based on individual patient characteristics, pose significant challenges to standardizing therapies in practice, rendering it practically unachievable. Moreover, the absence of a patentable and commercializable drug further restricts funding opportunities for studies. However, the rise of concepts like real-world data (RWD) and real-world evidence (RWE) can facilitate the design and implementation of future clinical trials. Noninferiority studies and the utilization of synthetic or external control groups can bring investigations and evidence production closer to real-world scenarios [123]. Additionally, patient databases with predefined protocols, such as Data Biologics, offer the potential for real-time analysis of patients’ clinical outcomes.

This systematic review has some limitations, including the heterogeneity of studies since there are different etiologies of low back pain, different comparators and sites of PRP application, and the protocols used for PRP preparation. This heterogeneity makes it challenging to make direct comparisons between studies and may affect findings generalization. Additionally, long-term follow-up studies, including head-to-head comparative studies directly comparing PRP with other treatment modalities, will be important to assess the durability and sustained efficacy of PRP therapy for chronic low back pain. Finally, despite the promising results reported, the overall level of evidence found for PRP therapy for low back pain is moderate. In this context, conducting a meta-analysis for this systematic literature review was not feasible because of the heterogeneity, limited sample sizes, and varying follow-up durations among the available studies. The wide variability in these factors made it challenging to pool data and draw meaningful conclusions across the studies. However, a qualitative analysis and narrative synthesis of the evidence were performed to present a comprehensive overview of the available literature.

## 5. Conclusions

To the best of our knowledge, this systematic review represents one of the most comprehensive analyses of using PRP in the management of low back pain. Based on the collective findings of the included studies, we determined that the overall level of evidence supporting the use of PRP in low back pain is categorized as level II. Notably, the use of PRP in the lumbar spine has demonstrated a low incidence of adverse events when compared to similar spinal injection techniques, with well-documented safety profiles.

Large-scale, multicenter studies that encompass diverse patient populations are still needed to strengthen the current evidence. Further studies will be helpful to unveil the efficacy, optimal treatment protocols, and long-term outcomes associated with PRP therapy for low back pain. By addressing these research gaps, it will be possible to enhance our understanding of PRP’s potential as a valuable therapeutic option for individuals suffering from this debilitating condition.

## Figures and Tables

**Figure 1 biomedicines-11-02404-f001:**
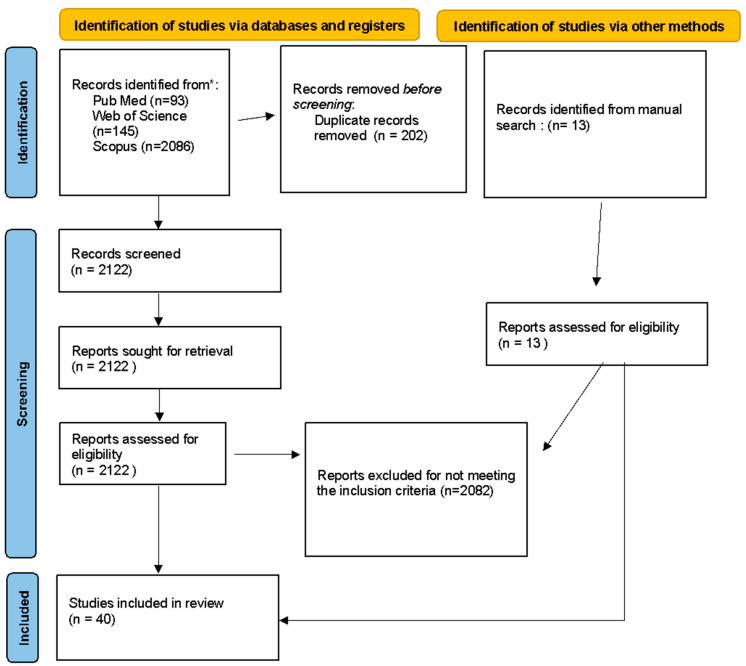
PRISMA flowchart.

**Figure 2 biomedicines-11-02404-f002:**
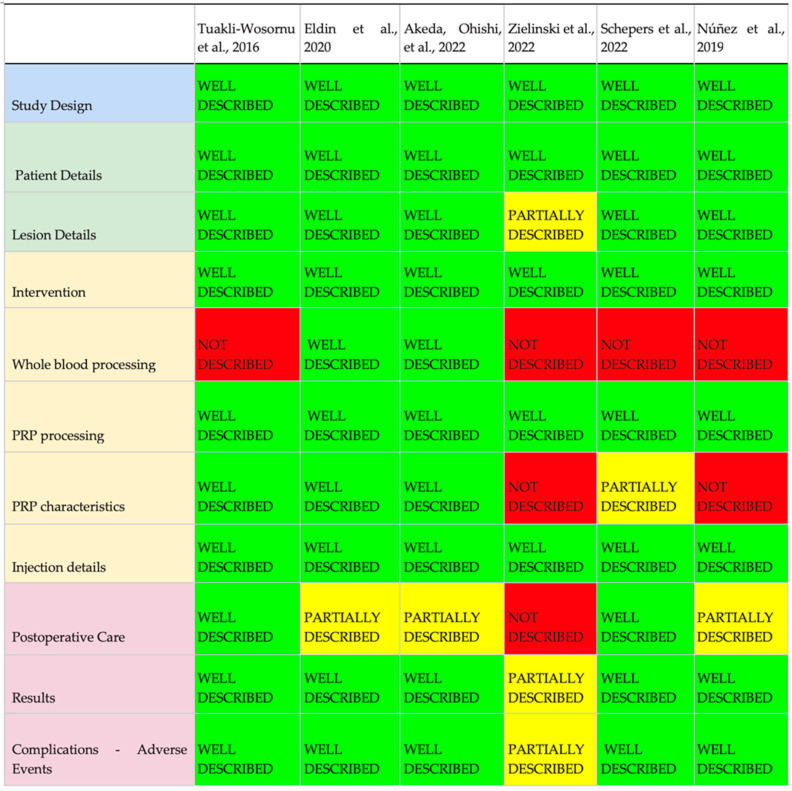
Intradiscal PRP—analysis of topics described in the study methodology, adapted from MIBO checklist [58,68,69,70,71,72].

**Figure 3 biomedicines-11-02404-f003:**
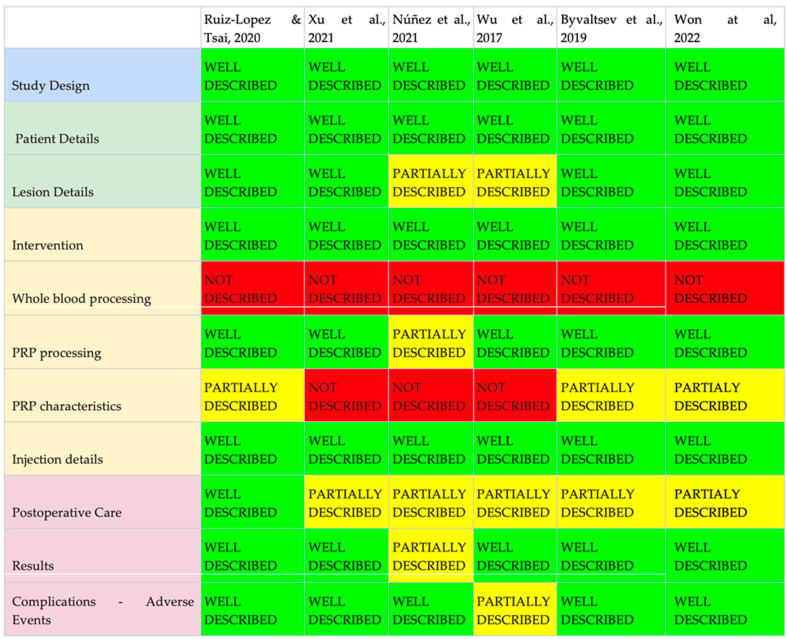
Epidural, facet, and muscle PRP—analysis of topics described in the study methodology, adapted from MIBO checklist [63,74,75,76,77,78].

**Table 1 biomedicines-11-02404-t001:** Summary of the selection criteria according PICO elements.

Criteria	Determinants
P (Population)	People with low back pain
I (Intervention)	PRP injection
C (Comparator)	Steroids, hyaluronic acid, ozone, saline, contrast medium
O (Outcome)	Pain control, less disability,
Study Design	RCTs, NRCTs, case series with more than 10 patients.

**Table 2 biomedicines-11-02404-t002:** Qualitative-modified approach to the grading of evidence [24].

Level I	Strong evidence obtained from multiple relevant high-quality randomized controlled trials for effectiveness.
Level II	Moderate evidence obtained from at least one relevant high quality randomized controlled trial or multiple relevant
	moderate or low-quality randomized controlled trials.
Level III	Fair evidence obtained from at least one relevant high quality nonrandomized trial or observational study with
	multiple moderate or low-quality observational studies.
Level IV	Limited Evidence obtained from multiple moderate or low quality relevant observational studies.
Level V	Consensus-based opinion or consensus of large group of clinicians and/or scientists for effectiveness, as well as to assess
	preventive measures, adverse consequences, and effectiveness of other measures.

**Table 3 biomedicines-11-02404-t003:** Randomized clinical trials—PRP on intervertebral disc.

	Author (Reference)	Comparator Control Group	PRP Processing	PRP Characteristics	Sample (*n*)	Adverse Event	Follow-Up
1	(Tuakli-Wosornu et al., 2016) [68]	placebo (contrast agent)	PRP Harvest Technol. Corporation (Plymouth, Ma) centrifuge	LR-PRP	47	Not related	12 months
2	(Eldin et al., 2020) [69]	PRF	Single spin 1000 rpm 6 min	LP-PRP	132	Not related	6 months
3	(Akeda, Ohishi, et al., 2022) [58]	betamethasone	PRP Relesate	PRP Rel	16	Pain: 1 case	12 months
4	(Zielinski et al., 2022) [70]	placebo (saline)	PUREPRP 1st Spin 3800 RPM 1.5 min 2nd Spin 3800 RPM 5 min	LP-PRP	36	Pain: 1 case	8 weeks
5	(Schepers et al., 2022) [71]	placebo (saline + kefazol)	SmartPReP Single spin 1000 RPM 15 min	LR-PRP	89	Discitis: 1 case	12 months
6	(Núñez et al., 2019) [72]	ozone	1st Spin 1200 rpm 8 min 2nd Spin 1200 rpm 8 min	NR	67	Vagal crisis: 2 cases	12 months
7	(Navani et al. 2023) [73]	placebo (saline)	Emcyte Pure PRP	LP-PRP	40	None	12 months

**Table 4 biomedicines-11-02404-t004:** Randomized clinical trials—PRP via epidural, facet, or muscle injection.

	Author (Reference)	Comparator Control Group	PRP Processing	PRP Processing	Sample (*n*)	Adverse Event	Follow-Up
1	(Ruiz-Lopez & Tsai, 2020) [74] Caudal Epidural	triamcinolone	Single spin 14 min 1568 g	LR-PRP	50	1 case pruritus	6 months
2	(Xu et al., 2021) [75] Transforaminal	betamethasone	1st Spin 1600 rpm 10 min 2nd Spin 3200 rpm 10 min	LP-PRP	124	Not related	12 months
3	(Núñez et al., 2021a) [76] Interlaminar	triamcinolone	Plasmaferesis—Plasma rich in growth factors	PRGF	93	1 case headache	12 months
4	(Wu et al., 2017) [63] Facet	betamethasone	PRP 1st Spin 200 g 10 min 2nd Spin 400 g 10 min	LR-PRP	46	Not related	6 months
5	(Byvaltsev et al., 2019) [77] Facet	hyaluronic acid	PRP Single spin 450 g 20 min	LP-PRP	144	Not related	12 months
6	(Won, Kim & Kim, 2022) [78] Muscle	lidocaine	Prosys system (Korea)	LR-PRP	30	Not related	6 months

**Table 5 biomedicines-11-02404-t005:** Nonrandomized clinical trials of the use of PRP in low back pain.

	Author (Reference)	PRP Processing	PRP Characteristics	Target	Sample (*n*)	Adverse Event	Follow-Up
1	Levi et al. 2015 [79]	Smartprep (Harvest)		Disc	22	Not related	6 months
2	Akeda et al. 2017 [56]	1st Spin 3000 g 15 min 2nd Spin 180 g 15 min	PRP	Disc	14	Not related	12 months
3	Navani et al. 2019 [34]	Emcyte Pure PRP system	PRP	Disc	14	Not related	18 months
4	Sevgili et al., 2020 [80]	GPS III Biomet	PRP	Disc	22	Not related	6 months
5	Jain et al., 2020 [81]	Double spin DrPRP kit (Dr PRP USA LLC)	LR-PRP	Disc	20	Not related	6 months
6	Lutz et al., 2022 [82]	Emcyte PurePRP II kit.	LR-PRP	Disc	37	1 discitis case	18 months
7	Zhang et al., 2022 [83]	Regen Laboratories SA Harvest Techonol. Corp	LP-PRP	Disc	31	1 case of discitis	48 weeks
8	Jose Correa et al., 2017 [85]	Not described	Not described	Epidural	70	Not related	3 months
9	Bhatia, 2016 [84]	Not described	Not described	Epidural	10	Not related	3 months
10	Centeno, 2017 [64]	Platelet lysate	Platelet lysate	Epidural	470	Headache, Dural lesion	24 months
11	Bise et al., 2020 [86]	Single spin 620 g 15 min	PRP	Epidural	60	Not related	6 weeks
12	Jose Correa et al., 2019 [87]	Not described	PRGF	Epidural	175	Not related	24 months
13	Viet-Thang Le 2022 [88]	1st Spin 1600 rpm 10 min 2nd Spin 3200 rpm 10 min	PRP	Epidural TFI	25	Not related	12 months
14	Wu et al., 2016 [89]	1st Spin 200 g 10 min 2nd Spin 400 g 10 min	PRP	Facet	19	Not related	3 months
15	Byvaltsev, 2019 [90]	Single spin 450 g 20 min	PRP	Facet	49	Not related	18 months
16	Byvaltsev, 2022 [91]	Single spin 450 g 20 min	PRP	Facet	41	Hematoma	18 months
17	Hussein and Hussein 2016 [61]	1st Spin 1500 rpm 15 min 2nd Spin 3000 rpm 20 min PLRP.	LR-PRP	Lumbar Muscle	104	Not related	24 months
18	Darrow et al., 2019 [92]	Not described	PRP	Lumbar Muscles	67	Not related	4 months

**Table 6 biomedicines-11-02404-t006:** Nonrandomized clinical trials: multitarget approach for low back pain.

	Author (Reference)	PRP Processing	PRP Characteristics	Sample (*n*)	Adverse Event	Follow-Up
1	Schwartz et al. 2013 [93]	Kit Closed System (PROTEAL ^®^)	LP-PRP	60	Headache	6 months
2	Kirchner, 2012 [94]	PRGF—Endoret ^®^	PRGF + O3	82	Not related	12 months
3	Kirchner, Anitua, 2016 [95]	PRGF—Endoret ^®^	PRGF	86	Headache	18 months
4	Cameron 2017 [96]	Not Reported	Not Reported	50	Not related	6 months
5	Machado et al., 2021 [10]	1st Spin 200 g 15 min 2nd Spin 1600 g 10 min	LR-PRP	46	Not related	12 months
6	Torres Morera et al., 2021 [97]	Not Reported	LR-PRP	24	Not Related	18 months
7	Kirchner et al., 2021 [98]	PRGF—Endoret ^®^	PRGF	47	Not Reported	48 weeks
8	Godek et al., 2022 [99]	Angel System ^®^—Arthrex	LR-PRP	91	Not related	3 months
9	Machado et al., 2022 [100]	1st Spin 200 g 12 min 2nd Spin 1600 g 8 min	LR-PRP	23	Not related	12 months

**Table 7 biomedicines-11-02404-t007:** RCTs—pain evaluation.

Author Year	Evaluation Method	Sample Size	PRP Group	Control Group	Follow-Up Time	PRP Group	Control Group
			Baseline Values (*)			Follow-Up Values (*)	
Tuakli-Wosornu et al., 2016 [68]	NRS	47	7.98 (1.56)	7.72 (1.53)	8 weeks	5.82 (2.33)	6.83 (2.33)
					12 months	5.86 (2.20)	-
Eldin et al., 2020 [69]	VAS		8.45 ± 0.59	8.34 ± 0.77	6 months	6.84 ± 1.58	4.95 ± 2.07
Akeda, Ohishi, et al., 2022 [58]	VAS	16	6.83 (1.33)	5.94 (1.24)	12 months	1.49 (2.47)	2.3 (2.37)
Schepers et al., 2022 [71]	NRS	89	6.29 (1.23)	6.02 (1.48)	12 months	5.3	5.1
Núñez et al., 2019 [72]	Lattinen	67	EVA > 5 in both groups		12 months	90% EVA less than 2	20% EVA less than 2
Ruiz-Lopez & Tsai, 2020 [74]	VAS	50	7.48 (1.12)	7.18 (0.95)	6 months	6.08 (0.99)	7.53 (0.60)
Xu et al., 2021 [75]	VAS	124	6.0 (6.0–7.25)	6.0 (5.0–7.0)	12 months	2.0 (1.0–3.0)	2.0 (1.0–3.0)
Núñez et al., 2021 [76]	Lattinen	93	8.5	8.5	12 months	1.5	6.5
Wu et al., 2017 [63]	VAS	46	7.09 (1.08)	6.74 (1.10)	6 months	2.7	4.5
Byvaltsev et al., 2019 [77]	VAS	144	6.85 (5.5–7.6)	6.6 (6.0–7.4)	18 months	1 (0.8–1.8)	1.7 (0.5–2.0)

* Absolute values of baseline and post procedures.

**Table 8 biomedicines-11-02404-t008:** RCTs—functional outcomes.

Author Year	Evaluation Method	Sample Size	PRP Group	Control Group	Follow-Up Time	PRP Group	Control Group
			Baseline Values (*)			Follow-up Values (*)	
Tuakli-Wosornu et al., 2016 [68]	FRI	47	51.47 (15.62)	45.37. (15.61)	8 weeks	37.99 (19.60)	44.45 (19.60)
					12 months	33.98 (20.35)	
Akeda, Ohishi, et al., 2022 [58]	ODI	16	36.0 ± 11.8	33.3 ± 11.6	12 months	−26.6 ± 14.8	−13.9 ± 9.7
	RMQ		8.6 ± 4.8	9.3 ± 4.7	12 months	−8.8 ± 5.0	−4.2 ± 4.5
Schepers et al., 2022 [71]	RMQ	89	12.63 (5.35)	13.42 (4.39)	12 month	9.6 (3.1)	10.1 (3.3)
Ruiz-Lopez & Tsai, 2020 [74]	SF-36	50	31.30 (20.80)	34.74 (18.42	6 months	59.74 (22.57)	35.42 (21.32)
Xu et al., 2021 [75]	ODI	124	35.0 (26.35–44.0)	27.0 (21.0–43.0)	12 months	19.0 (15.5–30.0)	20.0 (17.3–40.0)
Núñez et al., 2021 [76]	LATINEN	93	15.9	15.6	12 months	1.7	12.3
Wu et al., 2017 [63]	ODI	46	60.64 (10.84)	59.66 (10.35)	6 months	29.41 (7.76)	44.11 (7.30)
	RMQ		17.15 (3.13)	17.28 (2.27)	6 months	8.19 (3.48)	13.60 (2.90)
Byvaltsev et al., 2019 [77]	ODI	144	68 (53; 78)	66 (58; 75)	18 months	6.5 (2; 10)	14 (12; 20)
Won, Kim & Kim, 2022 [78]	ODI	30	32.7 ± 9.8	32.2 ± 10.5	6 months	16.1 ± 11.9	20.2 ± 7.9
	RMQ		12.2 ± 2.9	11.6 ± 3.9	6 months	4.0 ± 2.9	5.6 ± 3.2

(*) Absolute values of baseline and post procedures.

**Table 9 biomedicines-11-02404-t009:** Risk of bias RCT studies according to the RoB II tool.

Author/ Year	Randomization Process	Deviations from Intended Intervention	Missing Outcome Data	Measurement of Outcome	Selection of Reported Result	Overall Bias
Tuakli-Wosornu et al., 2016 [68]	Low Risk	Low Risk	Low Risk	Low Risk	Low Risk	Low Risk
Akeda et al., 2022 [69]	Low Risk	Low Risk	Low Risk	Low Risk	Low Risk	Low Risk
Schepers et al., 2022 [71]	Low Risk	Low Risk	Low Risk	Low Risk	Low Risk	Low Risk
Núñez et al., 2019 [72]	Low Risk	Low Risk	High Risk	High Risk	Some Concerns	High Risk
Ruiz-Lopez & Tsai, 2020 [74]	Low Risk	Low Risk	Low Risk	Low Risk	Low Risk	Low Risk
Xu et al., 2021 [75]	Low Risk	Low Risk	Low Risk	Low Risk	Low Risk	Low Risk
Núñez et al., 2021 [76]	Low Risk	High Risk	High Risk	High Risk	Some Concerns	High Risk
Wu et al., 2017 [63]	Low Risk	Low Risk	Low Risk	Low Risk	Low Risk	Low Risk
Byvaltsev, 2019 [77]	Low Risk	Low Risk	Low Risk	Low Risk	Low Risk	Low Risk
Won, Kim & Kim, 2022 [78]	Low Risk	Low Risk	Low Risk	Low Risk	Low Risk	Low Risk
Navani, 2023 [73]	Low Risk	Low Risk	Low Risk	Low Risk	Low Risk	Low Risk

## Data Availability

No new data were created in this study. Data sharing is not applicable to this article.

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
