# Peer review of "Systematic Review of Platelet-Rich Plasma for Low Back Pain"

_biomedicines, 2023, doi:10.3390/biomedicines11092404_

Round 1

Reviewer 1 Report

The ionztroduction is too detailed, contains several unnecesary parts, that could e punblished in a textbook, but not an artoicle. The detailed presentation of thers therateutic options could be left out (or better said, shortened - only PRP is interesting here).

The presentation of the included trials is not transparent enough. The comparators should be presented in the tables.

Description of the PRP technique is not at the right section.

In the last 2 columns of table 6 and 7 is not clear whether these are change values or absolute values (if the latter - what was the baseline?)

It is nozt discussed whether treated and control groups were homogenous. 

The discussion section lacks the synthetic approach, it should contain the conclusion of the authors on different therapeutic approaches.

Author Response

The introduction is too detailed, contains several unnecessary parts that could be published in a textbook, but not an article. The detailed presentation of their therapeutic options could be left out (or better said, shortened - only PRP is interesting here).

Thank you very much for your careful revision and excellent commentaries, which significantly contributed to improving our manuscript and enhancing its quality. As this review article is part of a dedicated pain issue, we made a deliberate decision to present a more in-depth and comprehensive review of the subject to provide readers with a robust knowledge base. Specifically, we focused on the aspect that the existing treatment options often address only the symptoms or lesions caused by the degenerative process, rather than targeting the underlying degenerative mechanisms themselves. By delving deeper into this aspect, we aimed to offer readers a more nuanced understanding of the context in which the current studies on PRP therapy for low back pain were conducted. Nevertheless, we agree with you, and we made the necessary adaptations and adjustments to the length of this section, ensuring it remains concise and relevant while maintaining its informative value.

The presentation of the included trials is not transparent enough. The comparators should be presented in the tables.

The PICO elements have been successfully incorporated into a table. Additionally, the comparator data has been seamlessly integrated into the respective table.

Description of the PRP technique is not at the right section.

We agree with you. This information has been changed.

In the last 2 columns of table 6 and 7 it is not clear whether these are change values or absolute values (if the latter - what was the baseline?)

The baseline values are displayed in the 4th and 5th columns for the PRP and control groups, respectively. The follow-up values are shown in the 7th and 8th columns. All of these values are absolute.

It is not discussed whether treated and control groups were homogenous. 

Inserted this commentary

The discussion section lacks the synthetic approach, it should contain the conclusion of the authors on different therapeutic approaches.

Upon carefully revisiting the paper, we acknowledge that the synthetic approach on different therapeutic approaches is indeed well described in the beginning of the discussion, page 19: “The etiology of the pain and the sites of application were varied, with majority of the studies about the treatment of discogenic pain with intradiscal biologics. Evidence has consistently demonstrated that PRP therapy offers a less invasive and safe alternative for the treatment of chronic low back pain. Notably, the use of PRP has shown significant benefits without substantial side effects or complications. This highlights the potential of PRP as a promising therapeutic approach that not only addresses chronic low back pain effectively but also minimizes the risks associated with more invasive treatment options.”

Reviewer 2 Report

·     The article by Edilson Silva Machado and colleagues addresses an important issue, the efficacy of PRP in back pain. It deserves to be published after some corrections and additional contributions.

·     Citations and references: please modify to adapt to the requirements of the Biomedicines journal style.

·     Please, check the levels and the different points, for example, 1.2.1.3 Surgery is inside if 1.2.1 Conservative measures… and  “PRP in Degenerative Spine Disease” has not numbers in the level.

·     LR PRP and LP PRP abbreviations should be LR-PRP and LP-PRP.

·     In table 4 is cited Correa 2019 and Correa et al 2016, but in the reference list only appears Correa 2017 , as reference #73. It would be useful to include the reference numbers in the citations od the tables… for example, Correa el al 2019 [reference number].

·     In table 5, the citation of Kirchner et at has not the year. I insist in the relevance of include the reference numbers in each row/study.

·     Table 6. The year is missing for Schepers et al.

·     In the discussion, please add cites to the paragraph of “Comparisons between diferente concentrations of PRP were performed: average… of 10x more platelets than the baseline level.” Additionally, the citation 85 in this paragraph it is not the accurate.

·     In the discussion, “Furthermore, the comparative clinical evaluation with PRP suggested better results in the PRF group75” the citation 75 is also not correct.

·     The results of MIBO are presented in the discussion section, with table 2 and 3. These are results, and should be included in the results section.

·     Please include a paragraph in the discussion with the limitations of this review.

·     Please comment on why a meta-analysis has not been carried out.

·     One additional comment. This reviewer is surprised by the high number of authors included in this systematic review: 16 authors. This must be very well justified.

·     The authors state that they comply with the PRISMA guidelines, but do not enclose the checklist as supplementary material.

Finally, the authors must check the accuracy of the reference numbers and the correspondent citations.

Minor editing of English language required

Author Response

The article by Edilson Silva Machado and colleagues addresses an important issue, the efficacy of PRP in back pain. It deserves to be published after some corrections and additional contributions.

  • Citations and references: please modify to adapt to the requirements of the Biomedicines journal style.

 Thank you very much for your time reviewing this manuscript, your kind comments, and your valuable suggestions. Related to your request, we adapt it.

  • Please, check the levels and the different points, for example, 1.2.1.3 Surgery is inside if 1.2.1 Conservative measures… and  “PRP in Degenerative Spine Disease” has not numbers in the level.

We agree with you and we modified the text accordingly.

  • LR PRP and LP PRP abbreviations should be LR-PRP and LP-PRP.

We agree with you and we modified the text accordingly.

  • In table 4 is cited Correa 2019 and Correa et al 2016, but in the reference list only appears Correa 2017 , as reference #73. It would be useful to include the reference numbers in the citations od the tables… for example, Correa el al 2019 [reference number].

We agree with you and we modified the text accordingly.: citation numbers inserted in the tables. Correa papers: 73 (2016) and 76 (2019).

  • In table 5, the citation of Kirchner et at has not the year. I insist in the relevance of include the reference numbers in each row/study.

We agree with you and we modified the text accordingly.

  • Table 6. The year is missing for Schepers et al.

We agree with you and we modified the text accordingly.

  • In the discussion, please add cites to the paragraph of “Comparisons between diferente concentrations of PRP were performed: average… of 10x more platelets than the baseline level.” Additionally, the citation 85 in this paragraph it is not the accurate.

We agree with you and we modified the text accordingly: (70)

  • In the discussion, “Furthermore, the comparative clinical evaluation with PRP suggested better results in the PRF group75” the citation 75 is also not correct.

56 is the correct

  • The results of MIBO are presented in the discussion section, with table 2 and 3. These are results, and should be included in the results section.

We agree with you and we modified the text accordingly.

  • Please include a paragraph in the discussion with the limitations of this review.

We agree with you and we modified the text accordingly:

This systematic review has some limitations, including the heterogeneity of studies since there are different etiologies of low back pain, the sites of PRP application, and the protocols used for PRP preparation. This heterogeneity makes it challenging to make direct comparisons between studies and may affect findings generalization.

Additionally, long-term follow-up studies, including head-to-head comparative studies directly comparing PRP with other treatment modalities, will be important to assess the durability and sustained efficacy of PRP therapy for chronic low back pain. Finally, despite the promising results reported, the overall level of evidence found for PRP therapy for low back pain is moderate.

  • Please comment on why a meta-analysis has not been carried out.

It is now also included in the discussion:

In this context, conducting a meta-analysis for this systematic literature review was not feasible due to the heterogeneity, limited sample sizes, and varying follow-up durations among the available studies. The wide variability in these factors made it challenging to pool data and draw meaningful conclusions across the studies. However, a qualitative analysis and narrative synthesis of the evidence were performed to present a comprehensive overview of the available literature

  • One additional comment. This reviewer is surprised by the high number of authors included in this systematic review: 16 authors. This must be very well justified.

Due to the comprehensive nature of the study, in order to cover different aspects of PRP therapy for low back pain, multiple authors were involved to ensure a well-structured, reliable, and robust analysis of the literature. The collaboration of multiple authors ensured a thorough analysis of the literature, reducing the risk of bias and errors. Additionally, involving a larger team enhanced transparency, reproducibility, and credibility of the review. Finally, the inclusion of authors in this systematic review was justified, as they form an international network (Brazil, USA, and Europe) planning new studies in the area of PRP therapy for low back pain, leveraging diverse expertise and resources to advance research in regenerative medicine and spine care. All the participants actively contributed to this comprehensive study; however, to align with the journal's recommendations, the authorship was reduced from 16 to 11, comprising those who made the most significant contributions. The remaining five contributors will be acknowledged in the acknowledgement section for their valuable support and assistance.

  • The authors state that they comply with the PRISMA guidelines, but do not enclose the checklist as supplementary material.

We agree with you and we modified the text accordingly.

Finally, the authors must check the accuracy of the reference numbers and the correspondent citations.

It was revised and corrected.

Round 2

Reviewer 2 Report

Two tips to complete the review:

- please add citation numbers in the authors un ALL tables.

- The prisma checklist is missing. Please add It as suppl. material.

Ok

Author Response

16 August 2023

Dear Prof. Dr. Ms. Adelia Zhang

Assistant Editor of Biomedicines,

Re: biomedicines-2528696 – 2nd Revision

Thank you for the comments regarding our paper “Systematic Review of Platelet Rich Plasma for Low Back Pain”. We have reviewed the manuscript according to the critiques raised by the reviewers and thank them for their comments.

We hope you will consider the revised version of the manuscript suitable for publication in Biomedicines. All alterations in the manuscript are easily recognizable. A point-by-point answer to the issues raised by reviewers can be found below.

Reviewer 2

- please add citation numbers in the authors un ALL tables.

Thank you very much for your careful re-revision and tips that we agree. We added the citation numbers in the authors for all tables.

- The prisma checklist is missing. Please add It as suppl. material.

We added the PRISMA checklist.